| Biotechnology | Research Article

# GenomicGapID: leveraging spatial distribution of conserved genomic sites for broad-spectrum microbial identification

Vishwaratn Asthana,[1] Pallavi Bugga,[2] Clara Elaine Smith,[2] Catherine Wellman,[2] Zachary Dwight,[3] Piyush Ranjan,[4] Erika Martínez Nieves,[2] Robert P. Dickson,[4,5,6] J. Scott VanEpps[2,6,7,8,9]

**ABSTRACT** Bacterial detection and identification methods can be broadly classified as either untargeted with expansive taxonomic coverage or targeted with narrow taxonomic focus. Untargeted approaches, such as culture and sequencing, are often time-consuming and/or costly, whereas targeted methods, such as PCR, can offer faster and more cost-effective results but require *a priori* knowledge of the likely pathogen to select the appropriate assay. GenomicGapID, a novel approach that leverages the spatial distribution of conserved genetic regions across microbial genomes, represents a significant advancement in the field of microbial identification. This technique has the potential to provide the taxonomic breadth of culture and sequencing while maintaining the speed, simplicity, and cost-effectiveness of PCR. By leveraging the conservation and relative positioning of highly conserved coding regions across different species, GenomicGapID enables the development of universal primer sets that amplify the non-conserved gaps between these regions. This creates a unique electrophoretic signature that facilitates rapid and accurate target-agnostic microbial identification. In this study, we apply the principles of GenomicGapID to the critical task of identifying clinical pathogens. We focus on expanding the coverage of a previously developed universal bacterial identification system, which initially targeted the 16s–23s internal transcribed spacer (ITS) region and was capable of discerning 45 pathogens. To enhance this system, we assembled a comprehensive database of 189 clinically relevant bacterial species. We then identified conserved primer binding sites that produce unique amplicon size signatures for each species. While we found that the use of amplicon size signatures alone would require an impractical number of universal primer sets, we demonstrate that this challenge can be effectively mitigated through concurrent melt analysis. Ultimately, we show that just three universal primer sets, guided by the GenomicGapID framework, are sufficient to cover 189 clinical bacterial pathogens, representing a majority of microbes identified in positive cultures in a clinical microbiology setting, with experimental validation of a subset of these pathogens. This study not only enhances the existing universal bacterial identification system but also establishes GenomicGapID as a versatile and powerful tool in microbial diagnostics and beyond, paving the way for new avenues of research in genomics with the potential to advance molecular biology, clinical practice, and public health.

**IMPORTANCE** Rapid and accurate microbial identification is critical in both clinical and research settings. Traditional untargeted methods, such as culture and sequencing, are often time-consuming and expensive, while targeted techniques like PCR offer speed and cost-effectiveness but require pre-selection of pathogens. Our work introduces GenomicGapID, a novel bacterial identification system that provides the taxonomic breadth of untargeted methods, coupled with the speed, simplicity, and affordability of targeted PCR-based techniques. By leveraging the gap between conserved genetic regions and analyzing the associated unique electrophoretic and melt analysis

Address correspondence to Vishwaratn Asthana, asthanav@med.umich.edu, or J. Scott VanEpps, jvane@med.umich.edu.

V.A. is the founder of Pyogenix, Inc. The work described herein is covered by International Patent Application No. PCT/US2023/071182.

signatures, GenomicGapID enables target-agnostic bacterial identification using a parsimonious set of universal primers.

Our work has significant implications not only in clinical microbiology but also in genomics, environmental microbiology, and public health. We believe this manuscript aligns well with the mission of Microbiology Spectrum to publish innovative and impactful research that advances the field of microbial sciences.

**KEYWORDS**  genomics, PCR, microbial identification

Current bacterial detection and identification methods fall into two main categories: broad but untargeted techniques and narrow but targeted approaches. While broad-spectrum techniques such as culture and sequencing provide comprehensive taxonomic coverage, they are often slow and expensive. In contrast, targeted methods like polymerase chain reaction (PCR) are quicker and more cost-effective but require prior knowledge of the specific organism, limiting their utility without a strong hypothesis. In this study, we introduce GenomicGapID, a cutting-edge approach that bridges the gap between these methods. GenomicGapID harnesses the spatial distribution of conserved genetic regions across microbial genomes to offer a solution that is as taxonomically inclusive as culture and sequencing, yet as rapid and cost-efficient as PCR. By examining the organization and conservation of these genomic regions across a wide array of bacterial species, GenomicGapID enables the design of universal primer sets that amplify the variable regions between conserved sequences, providing a versatile tool for microbial identification. This creates a unique electrophoretic signature based on amplicon length that facilitates rapid and accurate, target-agnostic microbial identification. This can be coupled with concurrent melt analysis, which provides discriminatory power at the sequence level, to improve specificity, all without the need for bioinformatic analysis. Using these approaches, it is theoretically possible to identify a near unlimited range of microbes with a parsimonious set of PCR reactions (i.e., less than 5). A conceptual framework for GenomicGapID can be found in Fig. 1.

One particularly compelling application of GenomicGapID is the rapid and accurate identification of the broad range of microbial pathogens that afflict humans. Traditional culture-based methods, which are considered the gold standard for bacterial identification, are inherently time-consuming, often requiring 24 to 72 h to yield results (1). Alternative diagnostic technologies, such as multiplex polymerase chain reaction (PCR) and next-generation sequencing (NGS), are a promising alternative to culture-based methodologies; however, these technologies are often limited by factors, such as cost, time-to-result, and/or pathogen coverage (2–7). The development of a universal bacterial identification system has the potential to overcome these limitations by providing a rapid, cost-effective, and comprehensive approach to pathogen detection.

We have recently reported the development of such a universal bacterial identification system that utilizes a unique set of universal PCR primers targeting the internal transcribed spacer (ITS) regions between conserved bacterial genes (8). Amplification of these variable regions generates a distinct signature of amplicons with varying lengths for different bacterial species, enabling the rapid and target-agnostic differentiation of pathogens based on their genetic profiles. The initial iteration of this system, which focused on the 16s–23s ITS region, demonstrated the ability to reliably identify 45 clinically relevant bacterial species. However, the number of clinically relevant microbial pathogens is far greater than this initial panel, necessitating further expansion and refinement of the system. Using GenomicGapID, we (i) assemble a comprehensive database of 189 clinically relevant bacterial species (sufficient to identify >80% of all microbes identified via positive culture in a clinical microbiology setting); (ii) identify sufficiently conserved primer binding sites to facilitate universal primer binding across species; (iii) narrow down these conserved primer binding sites to those that produce a unique electrophoretic pattern for each species; (iv) incorporate concurrent melt analysis

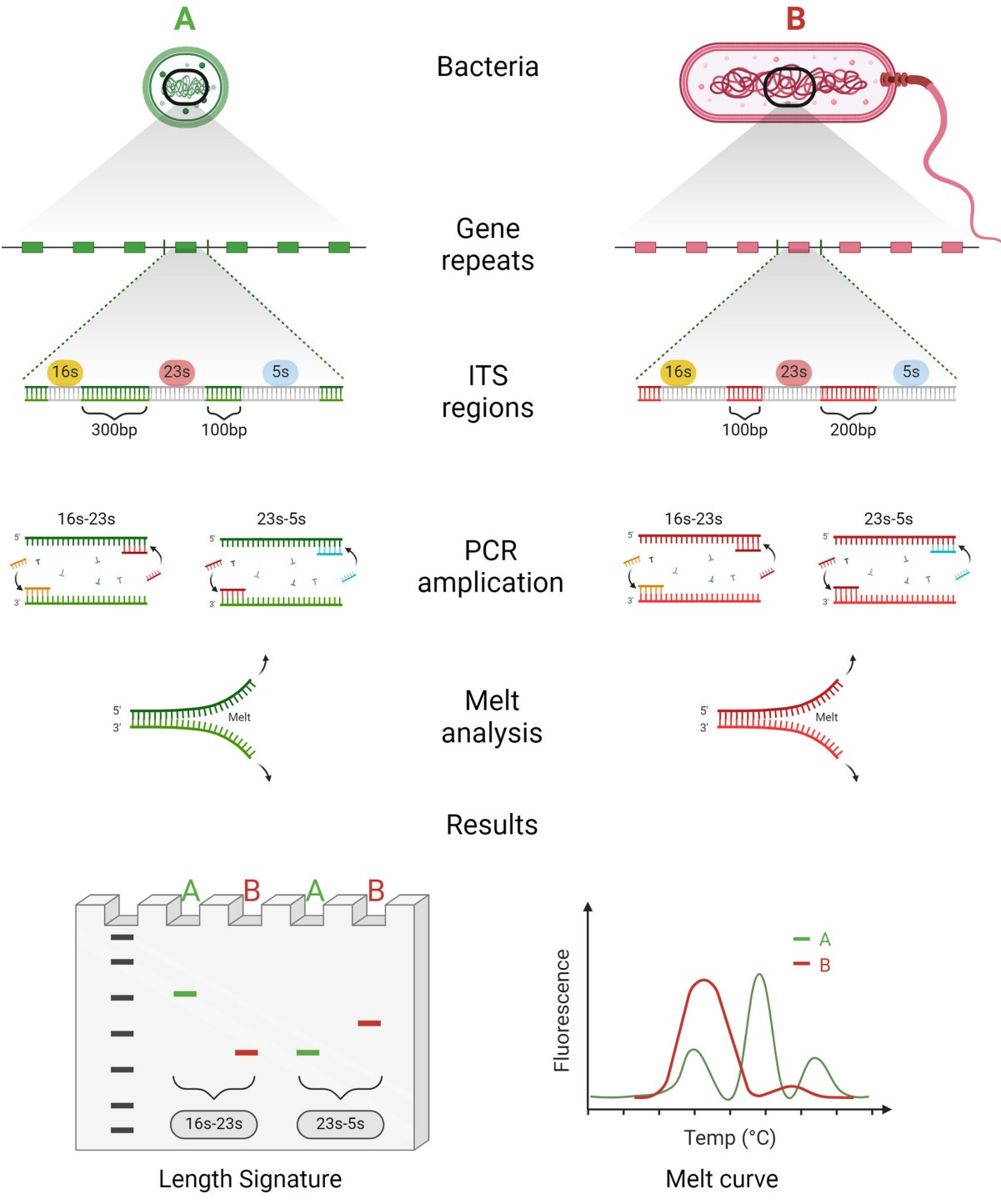

**FIG 1** Conceptualization of the universal bacterial identification system in the context of GenomicGapID. GenomicGapID explores the conservation and relative position of highly conserved coding regions in the microbial genome across space and species. Select genes are highly conserved across the bacterial kingdom. Leveraging the gaps between these variably spaced conserved regions using universal primers has the potential to open new lines of scientific investigation. For example, universal primers targeting the flanks of conserved bacterial genes (including the 16s, 23s, and 5s ribosomal segments) can be used to PCR amplify the heterogeneous ITS regions, producing a unique electrophoretic pattern based on the length of the ITS regions, as well as melt signature based on sequence composition, both of which in turn can be used to identify the bacteria of interest.

to differentiate species with an overlapping electrophoretic pattern; and (v) experimentally corroborate our *in silico* findings on a representative subset of the pathogen database.

This study not only enhances the existing universal bacterial identification system but also establishes GenomicGapID as a versatile and powerful tool in microbial diagnostics and beyond, with the potential to open new areas of investigation in genomics, with significant implications for molecular biology, clinical practice, and public health. A table comparing the GenomicGapID optimized universal bacterial identification system with existing modalities can be found in Table S1.

## MATERIALS AND METHODS

### Preparing bacterial genomic database and first-pass identification of conserved gene candidates

A literature search was performed to identify the most common bacterial pathogens that afflict humans with organisms pulled primarily from American and European databases. This list was corroborated with the internal clinical microbiology databases at Michigan Medicine. Together, a list of pathogens that would cover >80% of bacteria seen in positive cultures in a clinical microbiology laboratory was assembled, with inclusion of esoteric pathogens that are of particular clinical interest if identified. Care was taken to include pathogens for which an associated complete genome assembly was available on the National Center for Biotechnology Information (NCBI) microbial genome database. Complete genomes and associated annotation bacteria for all 189 bacteria were imported in FASTA format from NCBI. Highly conserved bacterial genomic targets, which would serve as potential sites for universal primer binding, were also pulled from the literature. These conserved genomic targets comprised rRNA, tRNA, and several essential genes, including *rpo, rpl, tuf, fus, lep, leu, rec, rps, dna, rnp, rpm,* and *gyr.* Annotation data for many bacteria in the database were incomplete. To address this, the *Escherichia coli* RefSeq file (U00096.3) was used as a reference. For each reportedly conserved gene of interest, its corresponding amino acid sequence in the *E. coli* annotation file was aligned with unidentified sequences of interest for every bacterium in the database. A sequence with >50% amino acid similarity was labeled a homologous gene (9).

### Simulating amplicon gaps between first-pass conserved gene candidates followed by calculation of unique electrophoretic signatures from amplicon lengths

Following annotation, the gap, measured in base pairs (bp), between all conserved genomic targets for all bacteria was calculated, using their 5′ or 3′ index position, to simulate the expected PCR amplicon length for each candidate gene combination. Calculated amplicons > 3 kb were removed from consideration given experimental constraints on polymerase extension. In addition, 400 bp was added to each simulated amplicon as an approximation for primer binding depth from the 3′ or 5′ end in the candidate gene (200 bp for each of the two conserved genes that flank the amplicon). To determine whether a bacterium can be uniquely identified based on the predicted amplicon profile generated by a given gene combination, the electrophoretic pattern for a given bacterium should (i) not overlap with the electrophoretic pattern of any other bacteria (within a 10% sizing error that would be expected from standard nucleic acid fragment analysis such as gel electrophoresis) or (ii) if the electrophoretic pattern overlaps, the bacteria may still be distinguishable if the number of repeats for a given amplicon size differs, creating an intensity difference for each band. After determining which gene combinations could uniquely identify which bacteria in the database, we iteratively combined each gene combination in order to maximize coverage of the database. Sequence conservation of a given gene combination across its intended bacterial targets was assessed using the multialign function on Matlab(R2022b).

## Second-pass narrowing of conserved gene candidates followed by combinatorial search to achieve complete coverage of the pathogen database

To adequately cover the entire pathogen database with a parsimonious set of primers, melt was incorporated (see subsection below). To determine the minimum number of gene combinations and associated primers required to cover the entire bacterial database using this revised methodology, a second-pass approach was undertaken. First, the list of conserved genes considered was dramatically reduced to only include candidates that could be bound with a limited set of primers. Instead of selecting genes with high conservation across the entire sequence, we selected genes with at least one contiguous and conserved 20 bp site allowing for the possibility of up to two mismatches, across at least 80 bacteria in the database. These sites would serve as potential primer binding regions. After pruning the list of conserved genes to the ribosomal subunits, a select group of tRNA, and the *tuf* gene using this approach, we repeated the above algorithmic search for gene combinations that would cover the entire bacterial database with the caveat that if a gene combination generates an amplicon for a bacteria, it would be considered uniquely identifiable using melt. Ultimately, the entire database could be covered by several permutations of three genes. The number of primers required to cover all permutations of the three gene candidates was calculated, and the combination that required the least number of primers was chosen. A generated primer was deemed able to bind to a gene if there were 16 matching bases with up to two mismatches.

### Incorporating melt simulation

Using melt analysis, the number of bacteria uniquely covered by a gene combination increases, such that any gene combination that generates an amplicon for a given bacteria can uniquely identify it by integrating both amplicon length and melt. Simulated melt profiles were generated using uMELT Quartz BATCH with the following input parameters: $Mg^{2+}$ = 1.5 mM, $Na^+$ = 50 mM, DMSO = 0%. When a given bacteria had multiple amplicons, a composite melt curve was generated by taking the average % helicity of all amplicons at a given $T_m$. To compare simulated and experimental melt, the tallest peak of the first derivative of the melt curve of a 200 bp reference strand (sequence below) was linearly shifted to align their simulated and experimental $T_m$. Next, the simulated and experimental melts for a bacterial target were linearly shifted to the same degree.

### Primer amplification simulation

As adapted from our prior work, a Matlab (R2022b) script was developed to establish primer binding sites and subsequent amplification profiles for each bacterial genome using the calculated thermodynamic affinities of the newly generated primers (8). In brief, the code searches each bacterial genome for primer binding sites by first probing for complementary base pair matches. This is followed by the calculation of the binding affinity ($\Delta G$) of each hit using integrated NUPACK code. Primer binding affinity was calculated at a temperature of 50°C, $Mg^{2+}$ of 1.5 mM, and $Na^+$ of 50 mM. Only primer binding sites with a $\Delta G < -10$ kcal/mol were found to be adequately stable to form a dimer, a prerequisite for amplification. Next, each primer is elongated 5′ - > 3′ until it reaches the next inversely oriented primer binding site along the genome, the intervening sequence of which is considered an amplicon. Only those amplicons <3,000 bp are included in the identity matrix generated for each universal ITS primer set.

### Cell culture and DNA extraction

Bacteria were grown overnight (12–18 h) at 37°C, at 200 rpm, in their respective media (Lysogeny broth [LB]) for *E. coli* (ATCC 25922); Tryptic soy broth with 1% glucose for *E. faecium* (ATCC BAA-2317). DNA was extracted using a *Lucigen* purification kit according to the manufacturer's instruction. DNA was reconstituted in TE buffer and stored at

−20°C. Purity and yields were quantified using a UV-Vis spectrophotometer (Thermo Scientific, Nanodrop 2000) and a Qubit fluorometer (ThermoFischer Scientific, Qubit 4 Fluorometer). Purified DNA was purchased for *C. jejuni* (ATCC 11168), *Leptospira interrogans* serovar Copenhageni (BAA-1198), and *Borrelia burgdorferi* (ATCC 35210) from the American Type Culture Collection (ATCC). The bacteria chosen represent different bacterial classes, including the Spirochaetia, Bacilli, Gammaproteobacteria, and Epsilonproteobacteria. *Leptospira interrogans* and *Borrelia burgdorferi* in particular are not covered by the 16s–23s universal primer pair and are not commonly encountered clinically, thus representing fringe cases for the system.

*E. coli* strains were purchased from ATCC. Deidentified clinical isolates were obtained from the Michigan Medicine Microbiology Laboratory. All variants were grown overnight (12–18 h) at 37°C, at 200 rpm, in LB. Overnights were diluted in PBS to $5 \times 10^7$ CFU/mL. Then, 600 µL of the diluted bacterial cell suspension was then mixed with 600 µL of BSA-treated ZR BashingBeads (0.1 and 0.5 mm) from Zymo and lysed using the Bullet Blender Storm 24 (NextAdvance) at speed 8 for 4 min. Then, 5 µL of the lysed cell suspension was then carried over to PCR and amplified using the protocol outlined below.

## Universal PCR amplification protocol

Bacterial DNA or lysed cell suspension was qPCR amplified using Phusion polymerase from NEB according to the manufacturer's instruction using the following thermocycling protocol: (i) 95°C for 3 min, (ii) 95°C for 15 s, (iii) 60°C for 30 s, (iv) 72°C for 1 min, and (v) repeat steps #2 to #4 for 35 cycles. Of note, the 16s–23s universal primer required a $T_m$ of 56°C instead of 60°C for optimal amplification. Amplification and melt were visualized using 1.5 µM of SYTO 9 intercalating dye on the Bio-Rad CFX Opus 96 Real-Time PCR System. Melt analysis was performed immediately following amplification using the following protocol: ramp up from 65°C to 95°C in 0.5°C increments with 5 s holds. A 200 bp reference strand (sequence information can be found in Table S2) was included with each melt run to help calibrate the signal. PCR reactants were subsequently separated via electrophoresis on a 10% polyacrylamide (PAGE) gel then imaged on a ChemiDoc XRS + molecular imager (Bio-Rad). Gel images were analyzed using GelAnalyzer 19.1 (www.gelanalyzer.com). Specifically, the intensity as a function of distance from the well was plotted for each lane of the gel. Peaks were identified by comparison to a standard 100 bp dsDNA ladder (New England Biolabs) that were fit to an exponential function of molecular weight versus distance.

## RESULTS

Prior to designing an optimal universal primer set that would provide broad microbial coverage, we first needed to construct a database of target microbes. Our primary focus was on human clinical pathogens; however, the approach described herein need not be limited to clinical medicine. Accordingly, we assembled a comprehensive database of the most common medically relevant bacterial pathogens, amounting to 189 bacteria in total (Fig. 2). To our knowledge, this is the most comprehensive pathogen panel outside of next generation sequencing (NGS).

Next, we imported the genomes for all 189 bacteria in the database, including associated annotation data, from NCBI. This was followed by a literature search for highly conserved bacterial genomic targets, which would serve as potential sites for universal primer binding. These first-pass conserved genomic targets comprised rRNA, tRNA, and several essential genes, including *rpo, rpl, tuf, fus, lep, leu, rec, rps, dna, rnp, rpm,* and *gyr* (10, 11). To determine how far we could get with amplicon length alone, the gap, measured in base pairs (bp), between all conserved genomic targets was calculated to establish which theoretical primer combination would generate the broadest and most discriminatory profile. The process by which a bacterium was determined to be uniquely covered by a universal primer pair is outlined in the Methods section.

This approach revealed diminishing returns with each new gene combination that asymptotically approaches 100% coverage (Fig. 3). For example, it takes two universal primer pairs to uniquely cover 97 bacteria or 51% of the database, four to cover 137 bacteria or 72%, and 10 to cover 170 bacteria or 90%. In addition to diminishing returns, many of the conserved genes derived from literature do not retain sufficient sequence homology across the bacterial database, making primer design difficult. As an example, the rplE gene, one of the most promising gene candidates with the highest coverage outside of the ribosomal genes, lacks five or more contiguous conserved bases across its intended bacterial targets (Fig. S1).

Given that amplicon length alone was not sufficient to adequately discriminate bacteria in the set, we sought to improve coverage of our existing 16s–23s universal primer pair, knowing that the 16s and 23s rRNA genes retain exceptionally high sequence homology across the bacterial pathogen database. On its own, the 16s–23s primer pair can uniquely discriminate 44 bacteria or 23% of the database based on amplicon length alone. However, the primer pair is able to generate an amplicon for the majority of bacteria in the database (175 bacteria or 93%), albeit with many bacteria generating overlapping electrophoretic patterns. To distinguish overlapping bacteria in this context, melt analysis was included. The melt properties of a strand are not only governed by its length but also its sequence composition, providing additional resolving power when the amplicon length signatures of multiple bacteria overlap. Using uMELT, a high-resolution melt simulation platform, it is possible to predict the melt profiles of the amplicons generated using each universal primer pair.

By incorporating melt analysis, the number of bacteria covered by the 16s–23s universal primer pair jumps to 175 bacteria or 93% of the database, which corresponds to every bacterium for which the 16s–23s primer pair is able to generate an amplicon for. The 16s and 23s rRNA genes are sufficiently conserved across the 175 bacteria that they can be bound with either a single primer pair or one that incorporates one to two redundant bases over the entire set. The remaining 14 bacteria or 7% of the database cannot be amplified using the 16s–23s primer set given the lengthy and unamplifiable gap between the 16s and 23s rRNA genes. A list of these bacteria can be found in Table S3. To find a primer pair that could reliably bind to the remaining 7% of the database and produce an amplicon in the target size range, we initially took a computationally intensive approach searching the entire genome space of all 14 bacteria, looking for two perfectly homologous 18 bp regions spaced <3 kb apart. While this method did not

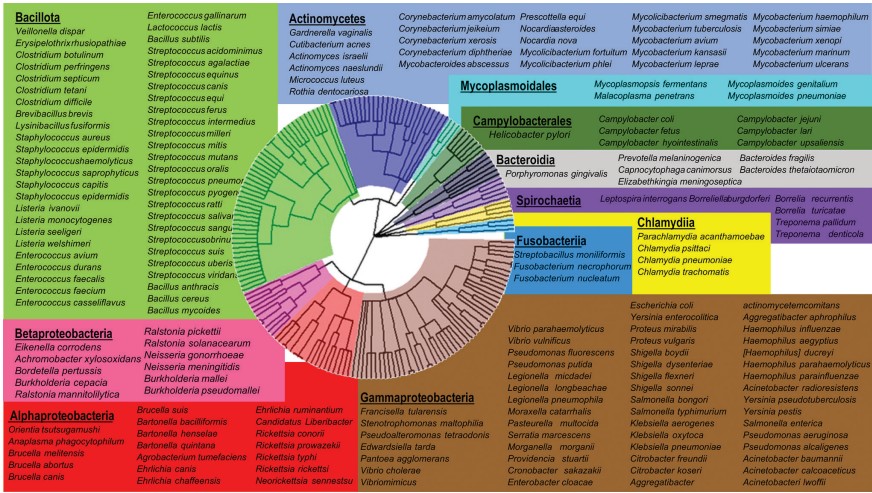

**FIG 2** Bacterial phylogenetic tree of clinically relevant pathogens. The following comprehensive list of bacteria represents the most common medically relevant bacterial pathogens, amounting to 189 bacteria in total. These pathogens represent the diagnostic target of the universal bacterial identification system and a viable application of GenomicGapID.

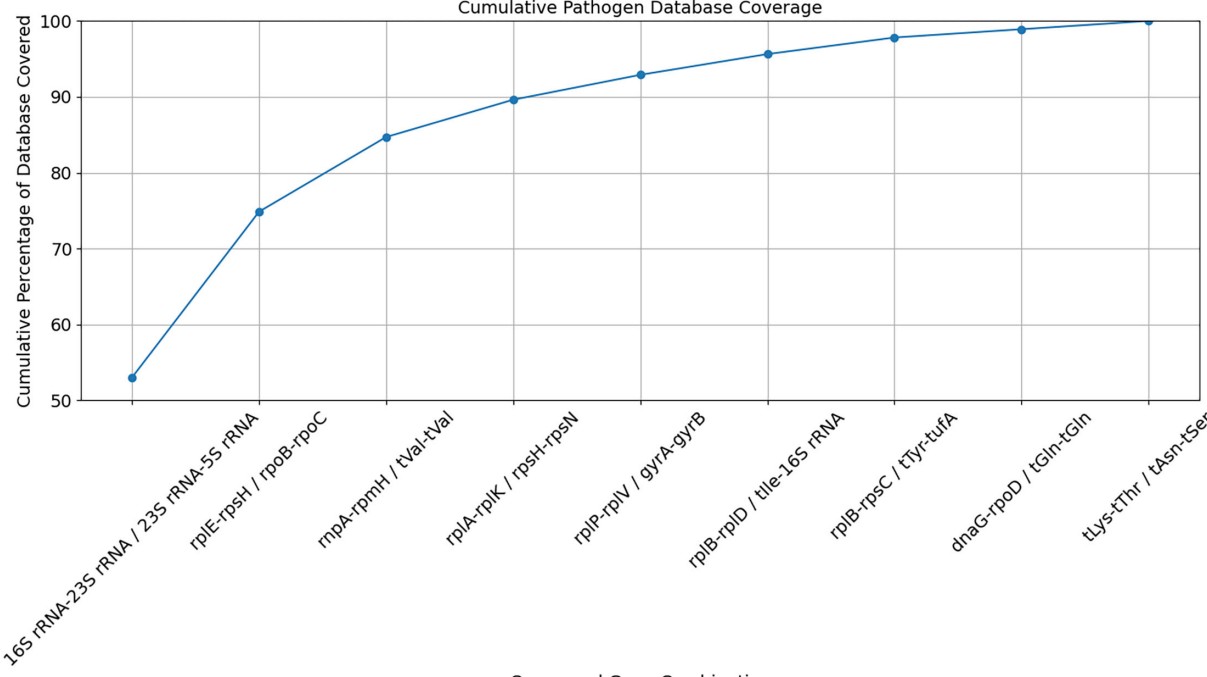

**FIG 3** Bacterial pathogen database coverage using an increasing number of universal primers targeting conserved gene regions. The gap between known conserved genomic targets comprising rRNA, tRNA, and several essential genes, including *rpo, rpl, tuf, fus, lep, leu, rec, rps, dna, rnp, rpm,* and *gyr* for 189 bacteria was calculated and incorporated into a combinatorial algorithm that calculates whether a bacterium can be uniquely identified based on the predicted amplicon length signature. After determining which gene combinations could uniquely identify which bacteria in the database, each gene combination was iteratively combined, two at a time, in order to maximize coverage of the database.

generate any candidates, the incorporation of two mismatches did reveal several targets; however, these regions of the genomes contained >90% "AT" sequence composition, making them unviable from a primer design standpoint (data not shown).

Given that one primer pair was not sufficient to cover the entire pathogen data set using both amplicon length and melt, a search for additional primers was performed with the goal of maintaining a minimalist set. Using a revised second-pass approach, we first searched for gene candidates with adequate sequence homology across the majority of the bacterial database. Instead of including genes with high conservation across the entire sequence, we selected genes with at least one contiguous 20 bp site and the possibility of up to two mismatches across at least 80 bacteria in the database. These sites would serve as potential primer binding regions. Then, 16s rRNA, 23s rRNA, and a select group of tRNAs were identified (*asp, thr, lys, met, val, gly, phe, leu, pro, ser, tyr, his, asn, cys, trp, arg, ile, ala,* and *glt*). *Ile, ala,* and *glt* were purposefully excluded as they are often found within the 16s–23s ITS region, which would lead to premature truncation of the 16s–23s amplicon if the primers were to be pooled. Of the protein-coding genes, only *tuf* was selected. While the 5s rRNA was not particularly well conserved across the entire database, it did maintain adequate conservation for the uncovered portion of the database.

Of these candidates, 23s–5s and *thr–tyr* not only covered the remaining 14 bacteria not covered by 16s–23s with the least number of redundant primers (see Supplementary Table S4 for the sequences of all universal primer pairs) but also generated redundant amplicons for a large portion of the bacterial database already covered by 16s–23s, providing an additional layer of specificity. In total, eight forward and reverse primers are required to cover the entire bacterial database (the 5s reverse primer required two additional variations to accommodate sequence heterogeneity at the target site).

The predicted amplicon length profile for all three universal primer pairs, calculated using thermodynamic affinities, and their corresponding simulated melt profiles can be found in Fig. 4A–D. A representative simulated melt profile for a set of bacteria with overlapping amplicon length signatures is shown in Fig. 4E. The numeric amplicon

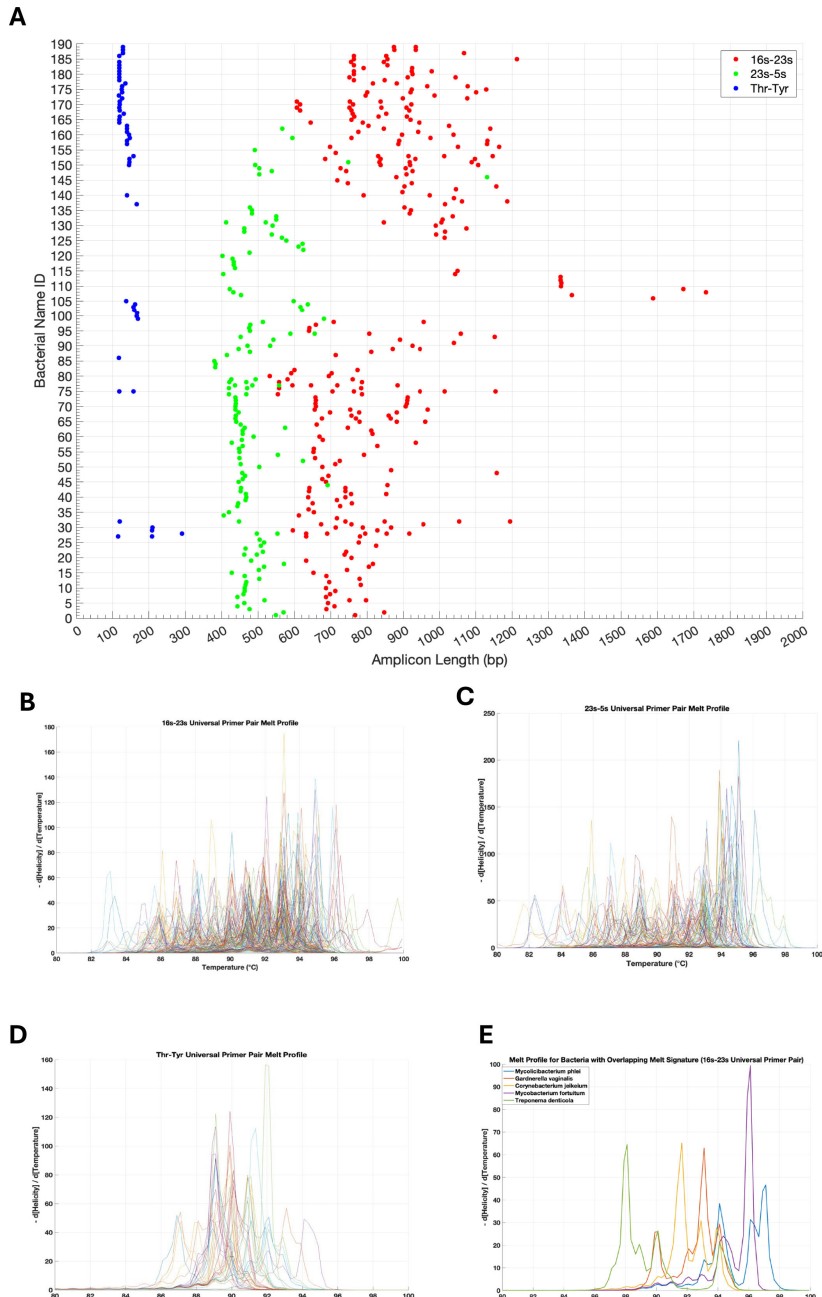

**FIG 4** Predicted amplicon length profile for all universal primer pairs and their corresponding simulated melt profiles. Primer binding and amplification were simulated using thermodynamic affinities. (A) Predicted amplicons for the 16s–23s, 23s–5s, and *thr–tyr* universal primer pairs were subsequently plotted. In the interest of space, bacterial names were omitted from the plot and can instead be found in the supplementary files along with the actual numeric amplicon lengths for each bacteria. Corresponding simulated melt profiles for the amplicons generated by the (B) 16s–23s, (C) 23s–5s, and (D) *thr–tyr* universal primer pairs were generated by inputting the amplicon sequences into uMELT. Bacterial IDs for these profiles were omitted in the interest of space. (E) Representative melt signatures for a cohort of bacteria whose amplicon length signatures overlap demonstrating resolvability.

length data and source amplicon sequences, the latter of which were used to generate the melt predictions, can be found in the supplementary files.

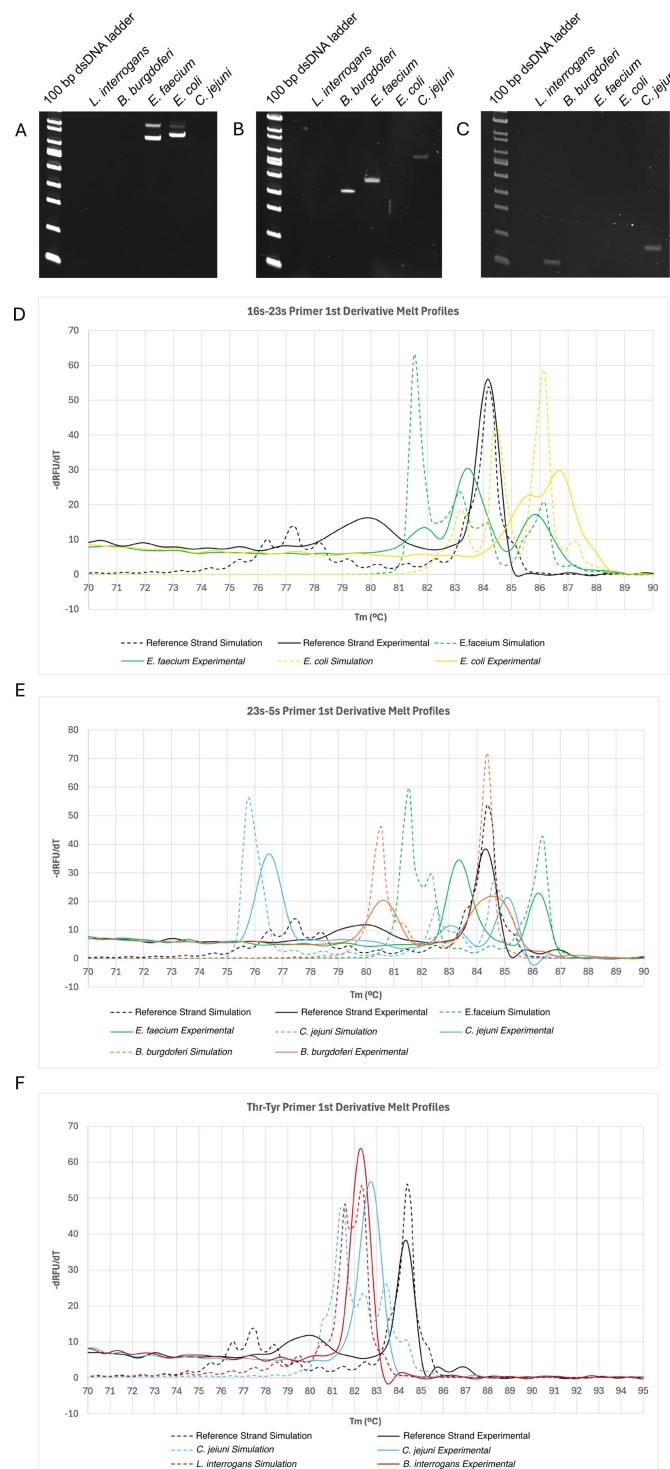

**FIG 5** Experimental validation of universal primer pairs. PAGE of PCR products from DNA isolated from different bacterial species amplified using the (A) 16s–23s, (B) 23s–5s, and (C) *thr–tyr* universal primer pairs. (D–F) Corresponding experimental melt data for amplified products from (A–C) respectively alongside simulated melt. A known synthetic reference strand is included to help align simulation and experimental data and normalize melt profiles between runs.

Experimental validation of the 16s–23s, 23s–5s, and *thr–tyr* universal primers on a representative and fringe cohort of bacteria can be found in Fig. 5 and Table 1. Of note, a 200 bp reference strand is included in all melt runs to help align simulation and experimental data and normalize melt profiles between runs. The reference strand helps account for deviations in salt concentration that would otherwise cause shifts along the $T_m$ axis as well as provide a reference for peak height when taking the irst derivative of RFU with respect to $T_m$.

## DISCUSSION

Sequence homology across essential genes is well preserved across the biological kingdom, including eubacteria, fungi, and protists (12). It may be possible then to explore the conservation and relative position of highly conserved coding regions within a kingdom across space and species, for applications such as organism identification, a term we coin GenomicGapID. A relatively simple tool to interrogate GenomicGapID is the use of universal primers to bind these conserved regions and amplify the intervening, non-conserved gaps, known in some applications as the ITS region, producing unique amplicons both in length and sequence composition. In the present study, we explore one such application of GenomicGapID, namely the development of an optimal universal primer set to reliably identify almost every known clinically relevant microbial pathogen.

To achieve this, we first constructed a comprehensive database of 189 medically significant bacteria. We imported genomic sequences and associated annotation data for all 189 bacteria from NCBI. This was followed by a thorough literature search to identify highly conserved bacterial genomic targets. These targets included rRNA, tRNA, and several essential genes. The gaps, measured in base pairs (bp), between these conserved genomic targets were calculated to determine which theoretical primer combinations would provide the broadest and most discriminatory profiles. Our combinatorial approach revealed that the 16s–23s and 23s–5s universal primer pairs covered a majority, albeit insufficient, portion of the database (i.e. able to uniquely identify a pathogen in the set) with rapidly diminishing returns for each subsequent universal primer pair. Additionally, many of the subsequent primer pairs relied on conserved genes extracted from the literature that ultimately lacked sufficient sequence homology across the microbial pathogen database for adequate primer design.

To enhance coverage, melt analysis was incorporated, which can orthogonally avail of sequence composition. We focused on improving the existing 16s–23s universal primer pair, which retains exceptionally high sequence homology across the bacterial pathogen database. The 16s–23s primer pair alone can uniquely discriminate 23% of the database based on amplicon length. However, incorporating high-resolution melt analysis (uMELT) significantly increases coverage to 93% by distinguishing overlapping electrophoretic patterns through their melt profiles, dictated in large part by the sequence of the amplicon. Melt has previously been utilized as a method of bacterial discrimination and identification; however, its utility as a standalone methodology is limited, particularly when the bacterial database is large, as the complexity and diversity of the bacterial genomic sequences can overwhelm the resolution capacity of high-resolution melt (13). When used in conjunction with amplicon length, however, high-resolution melt need only discriminate between 5 and 10 bacteria, significantly reducing the required discriminatory power of melt. Conveniently, melt analysis can be performed using a traditional qPCR apparatus, reducing both processing time and time to result.

Despite the incorporation of melt, 7% of the database remained unamplifiable using the 16s–23s primer set, either because of the extensive distance between the 16s and 23s genes or deviation from consensus at the primer binding site. We addressed this by repeating the process of searching for suitable conserved genomic targets, first computationally identifying genes with at least one contiguous 20 bp site and the possibility of up to two mismatches – a type of local conservation. This would ensure adequate universal primer pair binding. Despite significant overlap with literature, this revised, first-pass approach had a narrower scope of targets, including rRNA, select tRNA,

**TABLE 1** Predicted vs observed amplicon lengths in bp[a,b]

| Species | Predicted/observed 16s–23s | Predicted/observed 23s–5s | Predicted/ observed Thr–Tyr |
|---|---|---|---|
| *L. interrogans* | -/- | -/- | 117/122 |
| *B. burgdorferi* | -/- | 382/363 | -/- |
| *E. faecium* | 755, 852/725, 864 | 466/421 | -/- |
| *E. coli* | 763, 848/753, 854 | -/- | 118/- |
| *C. jejuni* | -/- | 621/594 | 159/157 |

[a]Predicted amplicons within a 10% tolerance are averaged together.
[b]- indicates no predicted and/or observed amplicons.

and the protein-coding *tuf* gene. Generally, genomic targets with local conservation also had high global sequence homology, but not vice versa.

Again, using a combinatorial approach, we sought the minimum number of universal primer pairs to cover the remainder of the database, settling on the 23s–5s rRNA and *thr–tyr* tRNA gaps. These new primer targets not only covered the previously unamplifiable bacteria but also provided redundant amplification for a portion of the database, enhancing overall specificity. We experimentally confirmed our amplicon length and melt simulations *in vitro* on a representative set of bacteria using gel electrophoresis and qPCR melt analysis.

Amplicon lengths largely demonstrated strong congruence between simulation and experimental results except for *thr–tyr* primer amplification of *E. coli*. We would expect a band at 118 bp; however, no band is visible. On closer inspection, there is a terminal 3′ mismatch in the *tyr* primer with the *E. coli* genome. Terminal 3′ mismatches are known to significantly impair PCR amplification efficiency, explaining the absence of an amplicon that would normally be expected using $\Delta G$ thresholds alone (14). Of note, the *tyr–tyr* primers were optimally designed for the microbes in the data set not covered by 16s–23s, explaining the suboptimal primer match with *E. coli*.

Experimental melt demonstrated reasonable alignment with simulation; however, it deviated to a greater extent than amplicon length. To correct for intraexperimental variability and facilitate alignment between simulation and experiment, a 200 bp reference strand was included in all melt runs, serving a somewhat similar function to a DNA ladder with gel electrophoresis. For visualization purposes, the tallest peak of the first derivative of the melt curve of the 200 bp reference strand was linearly shifted to align their simulated and experimental $T_m$. The simulated and experimental melts for a bacterial target were linearly shifted to the same degree. In general, we see reasonable overlap with peak number and position with some minor divergence. Linearly shifting the peak positions using the reference strand is likely not the optimal approach as the relationship between $T_m$ and $Na^+$ concentration is known to be nonlinear over a range of salt concentrations. This problem is exacerbated by the AT-rich regions that dominate the lower $T_m$ of the melt profile. In addition, simulated melts were calculated using the consensus sequences provided on NCBI. Minor strain level variations in sequence can lead to slight alterations in the melt profile also causing variableness (15–18).

Peak height and curve features are not as reliable when comparing simulated and experimental melt. It is likely that the curve features would more closely align if the melt ramp was allowed to proceed more gradually. In addition, peak height is more complicated as the relative contribution of each amplicon to the melt profile is presumed to be stoichiometric; however, we have observed that not all amplicons in a given bacteria amplify equally using the universal primer approach. This would shift the relative contribution and accordingly height of the pooled amplicons. Despite these issues, our approach appears to be a reasonable start.

Of note, another way to visualize the data is to take the difference of the percent helicity between the reference strand and target strand across all $T_m$ (Fig. S2). In general, melt is far more difficult to predict than primer binding as simulation predicts the graded

denaturation of often contiguous but sometimes disparate regions of a duplex, while primer binding and amplification can be approximated as a largely binary event (19, 20). Despite this, it appears that the experimental melts are consistent between strains and clinical isolates, paving the way for the construction of an experimental melt database based on real-world data to improve alignment and identification (Fig. S3). Not every bacterium would need an experimentally derived melt, though. There are several ways to approach this problem; however, the simplest would be to apply the equivalent of a sizing error to the simulated melt curves and determine which bacteria overlap both in amplicon length and melt. The melt profile for those specific bacteria that overlap can then be experimentally generated. Upon completion, the final identification algorithm would be run to confirm 100% specificity, even with the introduction of a sizing error into the simulated melt curves.

Overall, our data show that melt is a reliable tool to discriminate bacteria with overlapping amplicon length signatures using a tiered approach, where amplicon length among the three universal primers is used as a first-pass means of identification, that can be augmented by melt to improve specificity in equivocal cases. The redundancy in primers also helps.

Expansion of the universal bacterial identification system, a successful application of GenomicGapID, has significant implications for clinical diagnostics. By providing rapid and accurate identification of a comprehensive spectrum of bacterial pathogens, this system can improve patient outcomes by enabling timely and appropriate treatment without concern for false negatives in the setting of an inadequate diagnostic panel size. Of note, the system need not be limited to eubacteria. Other kingdoms of interest include the fungi and protists. Future research should aim to further refine primer designs to enhance specificity, such as accounting for terminal 3′ mismatches, and explore the application of this universal identification system in other settings, such as environmental and agricultural microbiology. GenomicGapID is certain to open new avenues for genomic research and diagnostic advancements.

## ACKNOWLEDGMENTS

This work was funded by the Frankel Innovation Initiative.

V.A., P.B., and J.S.V. were responsible for designing experiments. V.A., P.B., C.E.S., and C.W. performed experiments. V.A., P.B., J.S.V., P.R., and R.P.D. were responsible for analyzing data. Z.D. performed bulk melt analysis. V.A., P.B., E.M.N., Z.D., R.P.D., and J.S.V. wrote the manuscript.

During the preparation of this work, the authors used ChatGPT-4 in order to create a working outline for the Introduction and Discussion. After using this tool/service, the authors reviewed and edited the content as needed and take full responsibility for the content of the publication.

## AUTHOR AFFILIATIONS

[1]Department of Internal Medicine, Division of Hospital Medicine, University of Michigan, Ann Arbor, Michigan, USA

[2]Department of Emergency Medicine-Adult, University of Michigan, Ann Arbor, Michigan, USA

[3]Precision Biomarker Laboratories, Cedars-Sinai Medical Center, Beverly Hills, California, USA

[4]Division of Pulmonary and Critical Care Medicine, Department of Internal Medicine, University of Michigan Medical School, Ann Arbor, Michigan, USA

[5]Department of Microbiology and Immunology, University of Michigan Medical School, Ann Arbor, Michigan, USA

[6]Weil Institute for Critical Research and Innovation, University of Michigan, Ann Arbor, Michigan, USA

[7]Department of Biomedical Engineering, University of Michigan, Ann Arbor, Michigan, USA

[8]Program in Macromolecular Science and Engineering, University of Michigan, Ann Arbor, Michigan, USA

[9]Biointerfaces Institute, University of Michigan, Ann Arbor, Michigan, USA

## AUTHOR ORCIDs

Vishwaratn Asthana ⓘ http://orcid.org/0000-0002-2421-2820

J. Scott VanEpps ⓘ http://orcid.org/0000-0002-0805-0913

## ADDITIONAL FILES

The following material is available online.

### Supplemental Material

**Data S1 (Spectrum02817-24-S0001.xlsx).** 189 bacteria from the study with associated FASTA and RefSeq identifiers.
**Data S2 (Spectrum02817-24-S0002.xlsx).** Predicted amplicon lengths for all bacteria in the database for the 16s–23s, 23s–5s, and *thr–tyr* universal primers.
**Data S3 (Spectrum02817-24-S0003.xlsx).** Predicted melt sequences for all bacteria in the database for the 16s–23s, 23s–5s, and *thr–tyr* universal primers.
**Supplemental material (Spectrum02817-24-S0004.docx).** Fig. S1 to S3; Tables S1 to S4.

### Open Peer Review

**PEER REVIEW HISTORY (review-history.pdf).** An accounting of the reviewer comments and feedback.

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
