## [Reviewer comments · Microbiology Spectrum]

Microbiology Spectrum

GenomicGapID: Leveraging Spatial Distribution of Conserved Genomic Sites for Broad-Spectrum Microbial Identification

Vishwaratn Asthana, Pallavi Bugga, Clara Smith, Catherine Wellman, Zachary Dwight, Piyush Ranjan, Erika Nieves, Robert Dickson, and J. VanEpps

Corresponding Author(s): Vishwaratn Asthana, University of Michigan Michigan Medicine

Review Timeline:

Submission Date:	November 6, 2024
Editorial Decision:	December 17, 2024
Revision Received:	January 27, 2025
Accepted:	January 30, 2025

Editor: Mark Pandori

Reviewer(s): The reviewers have opted to remain anonymous.

Transaction Report:

DOI: <https://doi.org/10.1128/spectrum.02817-24>

Re: Spectrum02817-24 (GenomicGapID: Leveraging Spatial Distribution of Conserved Genomic Sites for Broad-Spectrum Microbial Identification)

Dear Dr. Vishwaratn Asthana:

Thank you for submitting your work for review. Below you will find comments, instructions from the Spectrum editorial office, and reviewer comments.

Revision Guidelines

Sincerely,
Mark Pandori
Editor
Microbiology Spectrum

Reviewer #1 (Comments for the Author):

Interesting approach in theory, im worried that in practice most species would not be easy to differentiate by the gel resolution. In particular closely related species, which are usually the clinically relevant (i.e. candida auris vs candida glabrata)

Reviewer #2 (Comments for the Author):

The manuscript "GenomicGapID: Leveraging Spatial Distribution of Conserved Genomic Sites for Broad-Spectrum Microbial Identification" by Asthana et al. uses a combined PCR and melt analysis technique to differentially identify clinically relevant bacteria. The technique is interesting and could save a significant amount of time and headache from sequencing.

My main comment is that this work is establishing a method for building a future database of PCR fragment size and melt curves, which does not yet exist as far as I could tell. Is there a plan to create this database or is any part of it already available (or planned to be made available after publication)?

While the predicted amplicon length seems to be in good agreement with the experimentally determined amplicon length, the predicted vs. observed melt analysis is more variable. This is addressed in the text but emphasizes the need for experimentally determined melt analysis rather than relying on prediction. Would every species of bacteria need to have this information present in the database? How would this be checked or validated?

Re: Spectrum02817-24 (“GenomicGapID: Leveraging Spatial Distribution of Conserved Genomic Sites for Broad-Spectrum Microbial Identification” by Asthana et al.)

Dear Dr. Mark Pandori,

Please convey our thanks to the reviewers for the careful examination of our manuscript. We believe the reviewer’s suggestions have greatly improved the strength and presentation of our investigation. Our responses to the reviewer’s comments, which are in bold, are indicated in the standard indented text below each comment.

Reviewer 1:

- 1) Interesting approach in theory, I’m worried that in practice most species would not be easy to differentiate by the gel resolution. In particular closely related species, which are usually the clinically relevant (i.e. candida auris vs candida glabrata)**

We appreciate the comment and the opportunity to clarify. While this is certainly a legitimate concern, we have found that closely related species are discernable using amplicon length alone. As an example, we have included the amplicon length profile for *Clostridium botulinum*, *Clostridium difficile*, and *Clostridium perfringens* below. Despite their proximity on the phylogenetic tree, the amplicon lengths of these related organisms are sufficiently discernable, even for just the 16s-23s universal primer set alone.

For those species which can’t be differentiated using amplicon length alone, one can use the melt signature, which as mentioned in the manuscript, is governed not only by the amplicon length, but also its sequence composition, providing adequate resolving power for all 189 bacteria (lines 291 – 302).

In response to the mention of differentiating *Candida auris* vs *Candida glabrata*, the system is not yet designed to identify fungi though this is an area of future investigation.

Reviewer 2:

- 2) My main comment is that this work is establishing a method for building a future database of PCR fragment size and melt curves, which does not yet exist as far as I could tell. Is there a plan to create this database or is any part of it already available (or planned to be made available after publication)?**

This is an excellent question. The presented database of simulated amplicon lengths and melt signatures provides adequate resolution to identify all 189 bacterial species, despite the minor variation in melt from simulation. While outside the scope of this manuscript, we are in the process of building out the database of experimentally derived amplicon lengths and melt signatures.

- 3) While the predicted amplicon length seems to be in good agreement with the experimentally determined amplicon length, the predicted vs. observed melt analysis is more variable. This is addressed in the text but emphasizes the need for experimentally determined melt analysis rather than relying on prediction. Would every species of bacteria need to have this information present in the database? How would this be checked or validated?**

Thank you for this comment. The experimentally determined melt adds confidence to the bacterial ID. Not every bacterium would need an experimentally derived melt though. There are several ways to approach this problem however the simplest would be to apply the equivalent of a sizing error to the simulated melt curves and determine which bacteria overlap both in amplicon length and melt. The melt profile for those specific bacteria that overlap can then be experimentally generated. Upon completion, the final identification algorithm would be run to confirm 100% specificity, even with the introduction of a sizing error into the simulated melt curves. This discussion has been added to the manuscript (lines 459-465).

Re: Spectrum02817-24R1 (GenomicGapID: Leveraging Spatial Distribution of Conserved Genomic Sites for Broad-Spectrum Microbial Identification)

Dear Dr. Vishwaratn Asthana:

Your manuscript has been accepted, and I am forwarding it to the ASM production staff for publication. Your paper will first be checked to make sure all elements meet the technical requirements. ASM staff will contact you if anything needs to be revised before copyediting and production can begin. Otherwise, you will be notified when your proofs are ready to be viewed.

Sincerely,
Mark Pandori
Editor
Microbiology Spectrum